# Norm matters: efficient and accurate normalization schemes in deep networks

**Elad Hoffer**[1]\*, **Ron Banner**[2]\*, **Itay Golan**[1]\*, **Daniel Soudry**[1]
{elad.hoffer, itaygolan, daniel.soudry}@gmail.com
{ron.banner}@intel.com

(1) Technion - Israel Institute of Technology, Haifa, Israel
(2) Intel - Artificial Intelligence Products Group (AIPG)

## Abstract

Over the past few years, Batch-Normalization has been commonly used in deep networks, allowing faster training and high performance for a wide variety of applications. However, the reasons behind its merits remained unanswered, with several shortcomings that hindered its use for certain tasks. In this work, we present a novel view on the purpose and function of normalization methods and weight-decay, as tools to decouple weights' norm from the underlying optimized objective. This property highlights the connection between practices such as normalization, weight decay and learning-rate adjustments. We suggest several alternatives to the widely used $L^2$ batch-norm, using normalization in $L^1$ and $L^\infty$ spaces that can substantially improve numerical stability in low-precision implementations as well as provide computational and memory benefits. We demonstrate that such methods enable the first batch-norm alternative to work for half-precision implementations. Finally, we suggest a modification to weight-normalization, which improves its performance on large-scale tasks. [2]

## 1 Introduction

Deep neural networks are known to benefit from normalization between consecutive layers. This was made noticeable with the introduction of Batch-Normalization (BN) [19], which normalizes the output of each layer to have zero mean and unit variance for each channel across the training batch. This idea was later developed to act across channels instead of the batch dimension in Layer-normalization [2] and improved in certain tasks with methods such as Batch-Renormalization [18], Instance-normalization [33] and Group-Normalization [38]. In addition, normalization methods are also applied to the layer parameters instead of their outputs. Methods such as Weight-Normalization [27], and Normalization-Propagation [1] targeted the layer weights by normalizing their per-channel norm to have a fixed value. Instead of explicit normalization, effort was also made to enable self-normalization by adapting activation function so that intermediate activations will converge towards zero-mean and unit variance [21].

### 1.1 Issues with current normalization methods

Batch-normalization, despite its merits, suffers from several issues, as pointed out by previous work [27, 18, 1]. These issues are not yet solved in current normalization methods.

**Interplay with other regularization mechanisms.** Batch normalization typically improves generalization performance and is therefore considered a regularization mechanism. Other regularization mechanisms are typically used in conjunction. For example, weight decay, also known as $L^2$ regularization, is a common method which adds a penalty proportional to the weights' norm. Weight decay was proven to improve generalization in various problems [24, 5, 4], but, so far, not for non-linear deep neural networks. There, [40] performed an extensive set of experiments on regularization and concluded that explicit regularization, such as weight decay, may improve generalization performance, but is neither necessary nor, by itself, sufficient for reducing generalization error. Therefore, it is not clear how weight decay interacts with BN, or if weight decay is even really necessary given that batch norm already constrains the output norms [16]).

**Task-specific limitations.** A key assumption in BN is the independence between samples appearing in each batch. While this assumption seems to hold for most convolutional networks used to classify images in conventional datasets, it falls short when employed in domains with strong correlations between samples, such as time-series prediction, reinforcement learning, and generative modeling. For example, BN requires modifications to work in recurrent networks [6], for which alternatives such as weight-normalization [27] and layer-normalization [2] were explicitly devised, without reaching the success and wide adoption of BN. Another example is Generative adversarial networks, which are also noted to suffer from the common form of BN. GAN training with BN proved unstable in some cases, decreasing the quality of the trained model [28]. Instead, it was replaced with virtual-BN [28], weight-norm [39] and spectral normalization [32]. Also, BN may be harmful even in plain classification tasks, when using unbalanced classes, or correlated instances. In addition, while BN is defined for the training phase of the models, it requires a running estimate for the evaluation phase – causing a noticeable difference between the two [19]. This shortcoming was addressed later by batch-renormalization [18], yet still requiring the original BN at the early steps of training.

**Computational costs.** From the computational perspective, BN is significant in modern neural networks, as it requires several floating point operations across the activation of the entire batch for every layer in the network. Previous analysis by Gitman & Ginsburg [11] measured BN to constitute up to $24\%$ of the computation time needed for the entire model. It is also not easily parallelized, as it is usually memory-bound on currently employed hardware. In addition, the operation requires saving the pre-normalized activations for back-propagation in the general case [26], thus using roughly twice the memory as a non-BN network in the training phase. Other methods, such as Weight-Normalization [27] have a much smaller computational cost but typically achieve significantly lower accuracy when used in large-scale tasks such as ImageNet [11].

**Numerical precision.** As the use of deep learning continues to evolve, the interest in low-precision training and inference increases [17, 36]. Optimized hardware was designed to leverage benefits of low-precision arithmetic and memory operations, with the promise of better, more efficient implementations [22]. Although most mathematical operations employed in neural-networks are known to be robust to low-precision and quantized values, the current normalization methods are notably not suited for these cases. As far as we know, this has remained an unanswered issue, with no suggested alternatives. Specifically, all normalization methods, including BN, use an $L^2$ normalization (variance computation) to control the activation scale for each layer. The operation requires a sum of power-of-two floating point variables, a square-root function, and a reciprocal operation. All of these require both high-precision to avoid zero variance, and a large range to avoid overflow when adding large numbers. This makes BN an operation that is not easily adapted to low-precision implementations. Using norm spaces other than $L^2$ can alleviate these problems, as we shall see later.

## 1.2 Contributions

In this paper we make the following contributions, to address the issues explained in the previous section:

- We find the mechanism through which weight decay before BN affects learning dynamics: we demonstrate that by adjusting the learning rate or normalization method we can exactly mimic the effect of weight decay on the learning dynamics. We suggest this happens since

certain normalization methods, such as a BN, disentangle the effect of weight vector norm on the following activation layers.

- We show that we can replace the standard $L^2$ BN with certain $L^1$ and $L^\infty$ based variations of BN, which do not harm accuracy (on CIFAR and ImageNet) and even somewhat improve training speed. Importantly, we demonstrate that such norms can work well with low precision (16bit), while $L^2$ does not. Notably, for these normalization schemes to work well, precise scale adjustment is required, which can be approximated analytically.

- We show that by bounding the norm in a weight-normalization scheme, we can significantly improve its performance in convnets (on ImageNet), and improve baseline performance in LSTMs (on WMT14 de-en). This method can alleviate several task-specific limitations of BN, and reduce its computational and memory costs (e.g., allowing to work with significantly larger batch sizes). Importantly, for the method to work well, we need to carefully choose the scale of the weights using the scale of the initialization.

Together, these findings emphasize that the learning dynamics in neural networks are very sensitive to the norms of the weights. Therefore, it is an important goal for future research to search for precise and theoretically justifiable methods to adjust the scale for these norms.

## 2 Consequences of the scale invariance of Batch-Normalization

When BN is applied after a linear layer, it is well known that the output is invariant to the channel weight vector norm. Specifically, denoting a channel weight vector with $w$ and $\hat{w} = w/\|w\|_2$, channel input as $x$ and $BN$ for batch-norm, we have

$$BN(\|w\|_2 \hat{w} x) = BN(\hat{w} x). \tag{1}$$

This invariance to the weight vector norm means that a BN applied after a layer renders its norm irrelevant to the inputs of consecutive layers. The same can be easily shown for the per-channel weights of a convolutional layer. The gradient in such case is scaled by $1/\|w\|_2$:

$$\frac{\partial BN(\|w\|_2 \hat{w} x)}{\partial(\|w\|_2 \hat{w})} = \frac{1}{\|w\|_2} \frac{\partial BN(\hat{w} x)}{\partial(\hat{w})}. \tag{2}$$

When a layer is rescaling invariant, the key feature of the weight vector is its direction.

During training, the weights are typically incremented through some variant of stochastic gradient descent, according to the gradient of the loss at mini-batch $t$, with learning rate $\eta$

$$w_{t+1} = w_t - \eta \nabla L_t (w_t) . \tag{3}$$

*Claim.* During training, the weight direction $\hat{w}_t = w_t/\|w_t\|_2$ is updated according to

$$\hat{w}_{t+1} = \hat{w}_t - \eta \|w_t\|^{-2} \left( I - \hat{w}_t \hat{w}_t^\top \right) \nabla L (\hat{w}_t) + O \left( \eta^2 \right)$$

*Proof.* Denote $\rho_t = \|w_t\|_2$. Note that, from eqs. 2 and 3 we have

$$\rho_{t+1}^2 = \rho_t^2 - 2\eta \hat{w}_t^\top \nabla L (\hat{w}_t) + \eta^2 \rho_t^{-2} \|\nabla L (\hat{w}_t)\|^2$$

and therefore

$$\rho_{t+1} = \rho_t \sqrt{1 - 2\eta \rho_t^{-2} \hat{w}_t^\top \nabla L (\hat{w}_t) + \eta^2 \rho_t^{-4} \|\nabla L (\hat{w}_t)\|^2}$$
$$= \rho_t - \eta \rho_t^{-1} \hat{w}_t^\top \nabla L (\hat{w}_t) + O \left( \eta^2 \right) .$$

Additionally, from eq. 3 we have

$$\rho_{t+1} \hat{w}_{t+1} = \rho_t \hat{w}_t - \eta \nabla L (\hat{w}_t \rho_t)$$

and therefore, from eq. 2,

$$\hat{w}_{t+1} = \frac{\rho_t}{\rho_{t+1}} \hat{w}_t - \eta \frac{1}{\rho_{t+1} \rho_t} \nabla L (\hat{w}_t)$$
$$= \left(1 + \eta \rho_t^{-2} \hat{w}_t^\top \nabla L (\hat{w}_t)\right) \hat{w}_t - \eta \rho_t^{-2} \nabla L (\hat{w}_t) + O \left( \eta^2 \right)$$
$$= \hat{w}_t - \eta \rho_t^{-2} \left( I - \hat{w}_t \hat{w}_t^\top \right) \nabla L (\hat{w}_t) + O \left( \eta^2 \right) ,$$

which proves the claim. □

Therefore, the step size of the weight direction is approximately proportional to

$$\hat{w}_{t+1} - \hat{w}_t \propto \frac{\eta}{\|w_t\|_2^2}. \tag{4}$$

in the case of linear layer followed by BN, and for small learning rate $\eta$. Note that a similar conclusion was reached by van Laarhoven [34], who implicitly assumed $\|w_{t+1}\| = \|w_t\|$, though this is only approximately true. Here we show this conclusion is still true without such an assumption. This analysis continues to hold for non-linear functions that do not affect scale, such as the commonly used ReLU function. In addition, although stated for the case of vanilla SGD, similar argument can be made for adaptive methods such as Adagrad [9] or Adam [20].

## 3   Connection between weight-decay, learning rate and normalization

We claim that when using batch-norm (BN), weight decay (WD) improves optimization only by fixing the norm to a small range of values, leading to a more stable step size for the weight direction ("effective step size"). Fixing the norm allows better control over the effective step size through the learning rate $\eta$. Without WD, the norm grows unbounded [31], resulting in a decreased effective step size, although the learning rate hyper-parameter remains unchanged.

We show empirically that the accuracy gained by using WD can be achieved without it, only by adjusting the learning rate. Given statistics on norms of each channel from a training with WD and BN, similar results can be achieved without WD by mimicking the effective step size using the following correction on the learning rate:

$$\hat{\eta}_{\text{Correction}} = \eta \frac{\|w\|_2^2}{\|w_{[\text{WD on}]}\|_2^2} \tag{5}$$

where $w$ is the weights' vector of a single channel, and $w_{[\text{WD on}]}$ is the weights' vector of the corresponding channel in a training with WD. This correction requires access to the norms of a training with WD, hence it is not a practical method to replace WD but just a tool to demonstrate our claim on the connection between weights' norm, WD and step size.

We conducted multiple experiments on CIFAR-10 [23] to show this connection. Figure 1 reports the test accuracy during the training of all experiments. We were able to show that WD results can be mimicked with step size adjustments using the correction formula from Eq. 5. In another experiment, we replaced the learning rate scheduling with norm scheduling. To do so, after every gradient descent step we normalized the norm of each convolution layer channel to

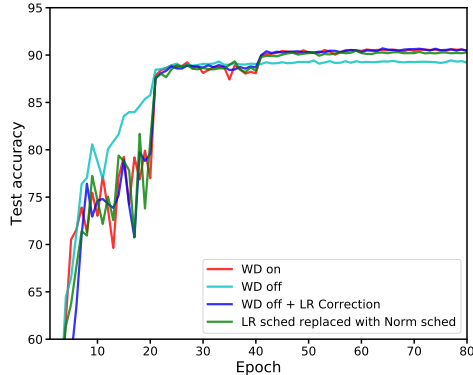

Figure 1: The connection between norm, effective step size and weight decay. *WD on*/*WD off* was trained with/without weight decay respectively. *WD off correction* was trained without weight decay but with LR correction as presented in Eq. 5. *LR sched replaced with Norm sched* is based on *WD on* norms but replacing LR scheduling with norm scheduling. (VGG11, CIFAR-10)

be the same as the norm of the corresponding channel in training with WD and keep the learning rate constant. When learning rate is multiplied by 0.1 in the WD training, we instead multiply the norm by $\sqrt{10}$, leading to an effective step size of $\frac{\eta}{\|W_{\text{WD on}}\|_2^2 \sqrt{10}^2} = 0.1 \frac{\eta}{\|W_{\text{WD on}}\|_2^2}$. As expected, when applying the correction on step-size or replacing learning rate scheduling with norm scheduling, the accuracy is similar to the training with WD throughout the learning process, suggesting that WD affects the training process only indirectly, by modulating the learning rate. Implementation details appear in supplementary material.

# 4 Alternative $L^p$ metrics for batch norm

We suggested above that the main function of BN is to neutralize the effect of the preceding layer's weights. If this hypothesis is true, then other operations might be able to replace BN, as long as they remain similarly scale invariant (as in eq. (1)) — and if we keep the same scale as BN. Following this reasoning, we next aim to replace the use of $L^2$ norm with scale-invariant alternatives which are more appealing computationally and for low-precision implementations.

Batch normalization aims at regularizing the input so that sum of deviations from the mean would be standardized according to the Euclidean $L^2$ norm metric. For a layer with $d-$dimensional input $x = (x^{(1)}, x^{(2)}, ..., x^{(d)})$, $L^2$ batch norm normalizes each dimension

$$\hat{x}^{(k)} = \frac{x^{(k)} - \mu^k}{\sqrt{\text{Var}[x^{(k)}]}} , \qquad (6)$$

where $\mu^k$ is the expectation over $x^{(k)}$, $n$ is the batch size and $\text{Var}[x^{(k)}] = \frac{1}{n}||x^{(k)} - \mu^k||_2^2$.

The computation toll induced by $\sqrt{\text{Var}[x^{(k)}]}$ is often significant with non-negligible overheads on memory and energy consumption. In addition, as the above variance computation involves sums of squares, the quantization of the $L_2$ batch norm for training on optimized hardware can lead to numerical instability as well as to arithmetic overflows when dealing with large values.

In this section, we suggest alternative $L^p$ metrics for BN. We focus on the $L^1$ and $L^\infty$ due to their appealing speed and memory computations. In our simulations, we were able to train models faster and with fewer GPUs using the above normalizations. Strikingly, by proper adjustments of these normalizations, we were able to train various complicated models without hurting the classification performance. We begin with the $L^1$-norm metric.

## 4.1 $L^1$ batch norm.

For a layer with $d-$dimensional input $x = (x^{(1)}, x^{(2)}, ..., x^{(d)})$, $L^1$ batch normalization normalize each dimension

$$\hat{x}^{(k)} = \frac{x^{(k)} - \mu^k}{C_{L_1} \cdot ||x^{(k)} - \mu^k||_1/n} \qquad (7)$$

where $\mu^k$ is the expectation over $x^{(k)}$, $n$ is the batch size and $C_{L_1} = \sqrt{\pi/2}$ is a normalization term.

Unlike traditional $L^2$ batch normalization that computes the *average squared deviation* from the mean (variance), $L^1$ batch normalization computes only the *average absolute deviation* from the mean. This has two major advantages. First, $L^1$ batch normalization eliminates the computational efforts required for the square and square root operations. Second, as the square of an $n$-bit number is generally of $2n$ bits, the absence of these square computations makes it much more suitable for low-precision training that has been recognized to drastically reduce memory size and power consumption on dedicated deep learning hardware [7].

As can be seen in equation 7, the $L^1$ batch normalization quantifies the variability with the normalized average absolute deviation $C_{L_1} \cdot ||x^{(k)} - \mu^k||_1/n$. To calculate an appropriate value for the constant $C_{L_1}$, we assume the input $x^{(k)}$ follows Gaussian distribution $N(\mu^k, \sigma^2)$. This is a common approximation (e.g., Soudry et al. [30]), based on the fact that the neural input $x^{(k)}$ is a sum of many inputs, so we expect it to be approximately Gaussian from the central limit theorem. In this case, $\hat{x}^{(k)} = (x^{(k)} - \mu^k)$ follows the distribution $N(0, \sigma^2)$. Therefore, for each example $\hat{x}_i^{(k)} \in \hat{x}^{(k)}$ it holds that $|\hat{x}_i^{(k)}|$ follows a half-normal distribution with expectation $E(|\hat{x}_i^{(k)}|) = \sigma \cdot \sqrt{2/\pi}$. Accordingly, the expected $L^1$ variability measure is related to the traditional standard deviation measure $\sigma$ normally used with batch normalization as follows:

$$E\left[\frac{C_{L_1}}{n} \cdot ||x^{(k)} - \mu^k||_1\right] = \frac{\sqrt{\pi/2}}{n} \cdot \sum_{i=1}^{n} E[|\hat{x}_i^{(k)}|] = \sigma.$$

Figure 2 presents the validation accuracy of ResNet-18 and ResNet-50 on ImageNet using $L^1$ and $L^2$ batch norms. While the use of $L_1$ batch norm is more efficient in terms of resource usage,

power, and speed, they both share the same classification accuracy. We additionally verified $L^1$ layer-normalization to work on Transformer architecture [35]. Using an $L^1$ layer-norm we achieved a final perplexity of 5.2 vs. 5.1 for original $L^2$ layer-norm using the base model on the WMT14 dataset.

We note the importance of $C_{L_1}$ to the performance of $L^1$ normalization method. For example, using $C_{L_1}$ helps the network to reach 20% validation error more than twice faster than an equivalent configuration without this normalization term. With $C_{L_1}$ the network converges at the same rate and to the same accuracy as $L^2$ batch norm. It is somewhat surprising that this constant can have such an impact on performance, considering the fact that it is so close to one ($C_{L_1} = \sqrt{\pi/2} \approx 1.25$). A demonstration of this effect can be found in the supplementary material (Figure 1).

We also note that the use of $L^1$ norm improved both running time and memory consumption for models we tested. These benefits can be attributed to the fact that absolute-value operation is computationally more efficient compared to the costly square and sqrt operations. Additionally, the derivative of $|x|$ is the operation $sign(x)$. Therefore, in order to compute the gradients, we only need to cache the sign of the values (not the actual values), allowing for substantial memory savings.

## 4.2   $L^\infty$ batch norm

Another alternative measure for variability that avoids the discussed limitations of the traditional $L^2$ batch norm is the *maximum absolute deviation*. For a layer with $d-$dimensional input $x = (x^{(1)}, x^{(2)}, ..., x^{(d)})$, $L^\infty$ batch normalization normalize each dimension

$$\hat{x}^{(k)} = \frac{x^{(k)} - \mu^k}{C_{L_\infty}(n) \cdot ||x^{(k)} - \mu^k||_\infty}, \tag{8}$$

where $\mu^k$ is the expectation over $x^{(k)}$, $n$ is batch size and $C_{L_\infty}(n)$ is computed similarly to $C_{L_1}(n)$ (derivation appears in appendix).

While normalizing according to the maximum absolute deviation offers a major performance advantage, we found it somewhat less robust to noise compared to $L^1$ and $L^2$ normalization.

By replacing the maximum absolute deviation with the mean of ten largest deviations, we were able to make normalization much more robust to outliers. Formally, let $s_n$ be the $n$-th largest value in $S$, we define $\text{Top}(k)$ as follows

$$\text{Top}(k) = \frac{1}{k} \sum_{n=1}^{k} |s_n|$$

Given a batch of size $n$, the notion of $\text{Top}(k)$ generalizes $L^1$ and $L^\infty$ metrics. Indeed, $L^\infty$ is precisely $\text{Top}(1)$ while $L^1$ is by definition equivalent to $\text{Top}(n)$. As we could not find a closed-form expression for the normalization term $C_{\text{Top}K}(n)$, we approximated it as a linear interpolation between $C_{L_1}$ and $C_{L_\infty}(n)$. As can be seen in figure 2, the use of $\text{Top}(10)$ was sufficient to close the gap to $L_2$ performance. For further details on $\text{Top}(10)$ implementation, see our code.

## 4.3   Batch norm at half precision

Due to numerical issues, prior attempts to train neural networks at low precision had to leave batch norm operations at full precision (float 32) as described by Micikevicius et al. [25], Das et al. [8], thus enabling only *mixed* precision training. This effectively means that low precision hardware still needs to support full precision data types. The sensitivity of BN to low precision operations can be attributed to both the numerical operations of square and square-root used, as well as the possible overflow of the sum of many large positive values. To overcome this overflow, we may further require a wide accumulator with full precision.

We provide evidence that by using $L^1$ arithmetic, batch normalization can also be quantized to half precision with no apparent effect on validation accuracy, as can be seen in figure 3. Using the standard $L^2$ BN in low-precision leads to overflow and significant quantization noise that quickly deteriorate the whole training process, while $L^1$ BN allows training with no visible loss of accuracy.

As far as we know, our work is the first to demonstrate a viable alternative to BN in half-precision accuracy. We also note that the usage of $L^\infty$ BN or its $\text{Top}(k)$ relaxation, may further help low-

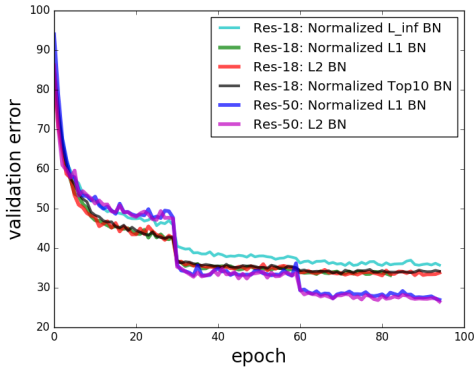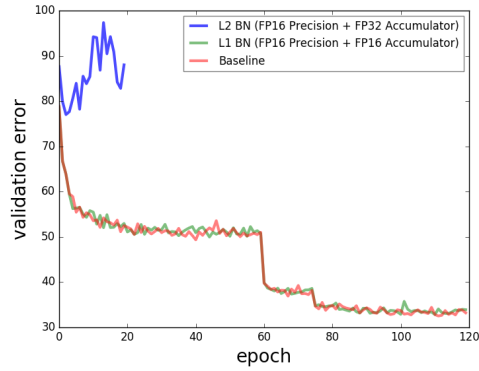

Figure 2: Classification error with $L^2$ batch norm (baseline) and $L^1$, $L^\infty$ and $\mathrm{Top}(10)$ alternatives for ResNet-18 and ResNet-50 on ImageNet. Compared to the baselines, $L^1$ and $\mathrm{Top}(10)$ normalizations reached similar final accuracy (difference < 0.2%), while $L^\infty$ had a lower accuracy, by 3%.

Figure 3: $L^1$ BN is more robust to quantization noise compared to $L^2$ BN as demonstrated for ResNet18 on ImageNet. The half precision run of $L^2$ BN was clearly diverging, even when done with a high precision accumulator, and we stopped the run before termination at epoch 20.

precision implementations by significantly lowering the extent of the reduction operation (as only $k$ numbers need to be summed).

## 5 Improving weight normalization

### 5.1 The advantages and disadvantages of weight normalization

Trying to address several of the limitations of BN, Salimans & Kingma [27] suggested weight normalization as its replacement. As weight-norm requires an $L^2$ normalization over the output channels of the *weight* matrix, it alleviates both computational and task-specific shortcomings of BN, ensuring no dependency on the current batch of sample activations within a layer.

While this alternative works well for small-scale problems, as demonstrated in the original work, it was noted by Gitman & Ginsburg [11] to fall short in large-scale usage. For example, in the ImageNet classification task, weight-norm exhibited unstable convergence and significantly lower performance (67% accuracy on ResNet50 vs. 75% for original).

An additional modification of weight-norm called "normalization propagation" [1] adds additional multiplicative and additive corrections to address the change of activation distribution introduced by the ReLU non-linearity used between layers in the network. These modifications are not trivially applied to architectures with complex structure elements such as residual connections [14].

So far, we've demonstrated that the key to the performance of normalization techniques lies in their property to neutralize the effect of weight's norm. Next, we will use this reasoning to overcome the shortcoming of weight-norm.

### 5.2 Norm bounded weight-normalization

We return to the original parametrization suggested for weight norm, for a given initialized weight matrix $V$ with $N$ output channels:

$$ w_i = g_i \frac{v_i}{\|v_i\|_2}, $$

where $w_i$ is a parameterized weight for the $i$th output channel, composed from an $L^2$ normalized vector $v_i$ and scalar $g_i$.

Weight-norm successfully normalized each output channel's weights to reside on the $L^2$ sphere. However, it allowed the weights scale to change freely through the scalar $g_i$. Following reasoning presented earlier in this work, we wish to make the weight's norm completely disjoint from its values.

We can achieve this by keeping the norm fixed as follows:

$$w_i = \rho \frac{v_i}{\|v_i\|_2},$$

where $\rho$ is a fixed scalar for each layer that is determined by its size (number of input and output channels). A simple choice for $\rho$ is by the initial norm of the weights, e.g $\rho = \|V\|_F^{(t=0)}/\sqrt{N}$, thus employing the various successful heuristics used to initialize modern networks [12, 13]. We also note that when using non-linearity with no scale sensitivity (e.g ReLU), these $\rho$ constants can be instead incorporated into only the final classifier's weights and biases throughout the network.

Previous works demonstrated that weight-normalized networks converge faster when augmented with mean only batch normalization. We follow this regime, although noting that similar final accuracy can be achieved without mean normalization but at the cost of slower convergence, or with the use of zero-mean preserving activation functions [10].

After this modification, we now find that weight-norm can be improved substantially, solving the stability issues for large-scale task observed by Gitman & Ginsburg [11] and achieving comparable accuracy (although still behind BN). Results on Imagenet using Resnet50 are described in Figure 4, using the original settings and training regime [14]. We believe the still apparent margin between the two methods can be further decreased using hyper-parameter tuning, such as a modified learning rate schedule.

It is also interesting to observe BWN's effect in recurrent networks, where BN is not easily applicable [6]. We compare weight-norm vs. the common implementation (with layer-norm) of an attention-based LSTM model on the WMT14 en-de translation task [3]. The model consists of 2 LSTM cells for both encoder and decoder, with an attention mechanism. We also compared BWN on the Transformer architecture [35] to replace layer-norm, again achieving comparable final performance (26.5 vs. 27.3 BLEU score on the original base model). Both sequence-to-sequence models were tested using beam-search decoding with a beam size of 4 and length penalty of 0.6. Additional results for BWN can be found in the supplementary material (Figure 2 and Table 1).

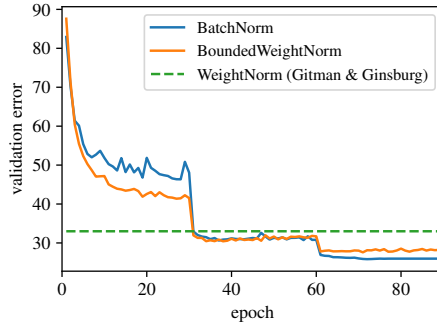

Figure 4: Comparison between batch-norm (BN), weight-norm (WN) and bounded-weight-norm (WN) on ResNet50, ImageNet. For weight-norm, we show the final results from [11]. Our implementation of WN here could not converge (similar convergence issues were reported by [11]). Final accuracy: BN - 75.3%, WN 67%, and BWN - 73.8%.

### 5.3 $L^p$ weight normalization

As we did for BN, we can consider weight-normalization over norms other than $L^2$ such that

$$w_i = \rho \frac{v_i}{\|v_i\|_p}, \quad \rho = \|V\|_p^{(t=0)}/N^{1/p},$$

where computing the constant $\rho$ over desired (vector) norm will ensure proper scaling that was required in the BN case. We find that similarly to BN, the $L^1$ norm can serve as an alternative to original $L^2$ weight-norm, where using $L^\infty$ cause a noticeable degradation when using its proper form (top-1 absolute maximum).

## 6 Discussion

In this work, we analyzed common normalization techniques used in deep learning models, with BN as their prime representative. We considered a novel perspective on the role of these methods, as tools to decouple the weights' norm from training objective. This perspective allowed us to re-evaluate the necessity of regularization methods such as weight decay, and to suggest new methods for normalization, targeting the computational, numerical and task-specific deficiencies of current techniques.

Specifically, we showed that the use of $L^1$ and $L^\infty$-based normalization schemes could provide similar results to the standard BN while allowing low-precision computation. Such methods can

be easily implemented and deployed to serve in current and future network architectures, low-precision devices. A similar $L^1$ normalization scheme to ours was recently introduced by Wu et al. [37], appearing in parallel to us (within a week). In contrast to Wu et al. [37], we found that the $C_{L_1}$ normalization constant is crucial for achieving the same performance as $L^2$ (see Figure 1 in supplementary). We additionally demonstrated the benefits of $L^1$ normalization: it allowed us to perform BN in half-precision floating-point, which was noted to fail in previous works [25, 8] and required full and mixed precision hardware.

Moreover, we suggested a bounded weight normalization method, which achieves improved results on large-scale tasks (ImageNet) and is nearly comparable with BN. Such a weight normalization scheme improves computational costs and can enable improved learning in tasks that were not suited for previous methods such as reinforcement-learning and temporal modeling.

We further suggest that insights gained from our findings can have an additional impact on the way neural networks are devised and trained. As previous works demonstrated, a strong connection appears between the batch-size used and the optimal learning rate regime [15, 29] and between the weight-decay factor and learning-rate [34]. We deepen this connection and suggest that all of these factors, including the effective norm (or temperature), are mutually affecting one another. It is plausible, given our results, that some (or all) of these hyper-parameters can be fixed given another, which can potentially ease the design and training of modern models.

## Acknowledgments

This research was supported by the Israel Science Foundation (grant No. 31/1031), and by the Taub foundation. A Titan Xp used for this research was donated by the NVIDIA Corporation.

## Footnotes

[2]Source code is available at `https://github.com/eladhoffer/norm_matters`

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
