[Supplementary Material]

# Norm matters: supplementary material

## 1   Implementation Details for weight-decay experiments

For all experiments, we used weight decay on the last layer with $\lambda = 0.0005$. The network architecture was VGG11 [4] with batch-norm after every convolution layer. Learning rate started from 0.1 and divided by 10 every 20 epochs (except for the norm scheduling experiment). The same random seed was used.

## 2   Importance of normalization constants

Figure 1 shows the scale adjustment $C_{L_1}$ is essential even for relatively "easy" data sets with small images such as CIFAR-10, and the use of smaller/bigger adjustments degrade classification accuracy.

Figure 1: *Left*: The importance of normalization term $C_{L_1}$ while training ResNet-56 on CIFAR-10. Without the use of $C_{L_1}$ the network convergence is slower and reaches a higher final validation error. We found it somewhat surprising that a constant so close to one ($C_{L_1} = \sqrt{\pi/2} \approx 1.25$) can have such an impact on performance. *Right*: We further demonstrate, with Res18 on ImageNet, that $C_{L_1} = \sqrt{\pi/2}$ is optimal: performance is only degraded if we modify $C_{L_1}$ to other nearby values.

### 2.1   Deriving $C_{L_\infty}$

As seen in main manuescript,

$$\hat{x}^{(k)} = \frac{x^{(k)} - \mu^k}{C_{L_\infty}(n) \cdot ||x^{(k)} - \mu^k||_\infty}, \tag{1}$$

To derive $C_{L_\infty}(n)$ we assume again the input $\{x_i\}_{i=1}^n$ to the normalization layer follows a Gaussian distribution $N(\mu^k, \sigma^2)$. Then, the maximum absolute deviation is bounded on expectation as follows

[3]:

$$\frac{\sigma \cdot \sqrt{\ln(n)}}{\sqrt{\pi \ln(2)}} \leq ||x^{(k)} - \mu^k||_\infty \leq \sigma\sqrt{2\ln(n)}.$$

Therefore, by multiplying the three sides of inequality with the normalization term $C_{L_\infty}(n)$, the $L^\infty$ batch norm in equation 1 approximates an expectation the original standard deviation measure $\sigma$ as follows:

$$l \leq C_{L_\infty}(n) \cdot ||x^{(k)} - \mu^k||_\infty \leq u$$

where $l = \frac{1+\sqrt{\pi \ln(4)}}{\sqrt{8\pi \ln(2)}} \cdot \sigma \approx 0.793\sigma$, and $u = \frac{1+\sqrt{\pi \ln(4)}}{2} \cdot \sigma \approx 1.543\sigma$.

## 3   Bounded-weight-norm experiments

Figure 2 depicts the impact of bounded-weight norm for the training of recurrent network on WMT14 de-en task. Additional results are summarized in Table 1.

Figure 2: Comparison between bounded weight-norm and baseline with no normalization in recurrent network training (LSTM attention-seq2seq network, WMT14 de-en)
.

Table 1: Results comparing baseline, $L^2$ based normalization with weight-norm (WN) by Salimans & Kingma [2] and our bounded-weight-norm (BWN)

| Network | Batch/Layer norm | WN | BWN |
|---|---|---|---|
| ResNet56 (Cifar10) | 93.03% | 92.5% | 92.88% |
| ResNet50 (ImageNet) | 75.3% | 67% [1] | 73.8% |
| Transformer (WMT14) | 27.3 BLEU | - | 26.5 BLEU |
| 2-layer LSTM (WMT14) | 21.5 BLEU | - | 21.2 BLEU |

Table 2: Results comparing baseline, and $L^1$ norm results (ppl for perplexity)

| Network | $L^2$ Batch/Layer norm | $L^1$ Batch/Layer norm |
|---|---|---|
| ResNet56 (Cifar10) | 93.03% | 93.07% |
| ResNet18 (ImageNet) | 69.8% | 69.74% |
| ResNet50 (ImageNet) | 75.3% | 75.32% |
| Transformer (WMT14) | 5.1 ppl | 5.2 ppl |