[Reviews · NeurIPS 2018]

Reviewer 1



This paper introduces and motivates L1 batch normalization (and variations: L_infinity batch-norm, etc.). With a theoretical study of the impact of standard (L2) batch normalization on the optimization process, the authors show that applying batch-norm makes the optimization weight-norm-invariant and is equivalent to just rescaling the learning rate (under the assumption that batch-norm is applied after a linear layer). Experiments confirm this. It is then proposed to use other kinds of normalization, namely L1, L_infinity, and a parameterized family between both, for computational efficiency reasons (faster to compute and requiring less precision, making it available for half-precision hardware), and it is shown that similar performance (in terms of accuracy) is reached as with the usual batch-norm (with experiments on ImageNet with ResNets). Finally, a similar study is conducted for weight normalization [Salimans & Kingma]. The difference with batch-norm is that the normalization is not the one of each neuronal activity over the mini-batch, but over of all input weights of each neuron together (independently of the mini-batch). As for batch-norm, It is proposed to decouple the norm of the weights from their direction, i.e. to use a representation of the kind x = rho u where u is unit-normed, for some Lp norm. Pros: - the paper is very well written. Not just easy to read, but very well motivated, with explanations, physical meaning and intuition, etc. The style of paper I'd like to see more often. - The topic of understanding and improving batch-norm is of prime interest. Significant insights are brought. - Experiments are meaningful. Remarks: - The analysis is specific to the case where batch-norm is applied after a linear layer. It seems this analysis should be still valid after ReLu, maxpoolings, etc. But what about after other activation functions? The experiments are performed with VGG and ResNets, which are conv-ReLu blocks. What about standard networks with really non-linear "non-linearities", i.e. not ReLu but maybe Elu, SeLu, etc.? Is the approach performance similar in practice for such non-"linear" networks? - Similarly, what about other gradients than the standard L2 one, for instance natural gradient, Adam, etc? as Adam is the most common optimizer now. Does the theoretical study still hold for such gradients? Any experiments with Adam? Actually, the paper does not state which optimizer is used, though I guess it might be SGD. - About notations: the paper is clear, but maybe some potential ambiguity in notations could be removed. For instance, sometimes the L2 norm is performed as a sum over all examples in the mini-batch; sometimes it is a sum of all input units for the considered neuron. Would it be possible to state explicitly the space over which the sum is performed? e.g. L2(mini-batch), L2(inputs). For instance line 160, or in equation 7, it is not straightforward to understand that the L2 norm stands for an expectation E_batch (and not on dimensions of x^(1) if any, e.g.). By the way, the expectation symbol would look better using \mathop{\mathbb{E}} for instance. Edit after authors' response: I would be interested in seeing the extension of the claim to momentum methods / Adam, as it is not straightforward and would be of high interest in the community. I keep the same score.

Reviewer 2



The paper presents an interesting view on the batch normalization method. In particular, the authors investigated the effect of a batch norm layer after a linear layer without any activation function. For this particular type of BN, the output is invariant to the scaling of the linear layer. On the other hand, in this case the gradient over the parameters with BN is scaled. The paper further derives that the step size of the “normalized” weights also gets scaled due to the BN layer. Based on the analysis, the authors show empirically by adjusting the learning rate accordingly, one can achieve similar performance without BN. Also the authors compared different norms for the BN layers. In all I like the paper. BN is used everywhere but little is known how and why it works. This paper provides an interesting angle toward understanding BN. Questions and Possible Issues: 1. I like the analysis in section 2. Still the proof in section 2 omitted some details. For example, line 107-108: taylor approximation is used without being mentioned. Similarly for line 109-110: another different form of taylor expansion is used without explanation. Could the author add more details to it? Or put it in appendix. 2. It appears to me the same analysis goes through for the Linear + RELU + BN, too. Correct me if I am wrong. 3. The lines in the figure 1, 2, and 3 are heavily overlapping with one another, making it extremely hard to tell which is which. Could the author try some other ways of visualization of the results? 4. Line 119 claims “with out WD, the norm grows unbounded, resulting in a decreased effective step size”. I don’t see why the norm can grow unbounded even without WD. Could the authors elaborate on it?

Reviewer 3



The paper presents three pieces of results regarding normalization techniques: 1) weight decay has an effect similar to modulating the learning rate; 2) L1 normalization works with low precision and achieves performance comparable to L2 batchnorm in high-precision settings; 3) fixing the scale of weight norm improves performance on large-scale data. The paper presents positive results on the performance of L1 normalization under low-precision setting, which might have potential impact. The experiments in Section 3 appear to be interesting and might potentially inspire future work. However, I have the following concerns. First, the L1 normalization experiments are weak. Under the high-precision setting, the authors claim that L1 normalization is faster than the standard L2 batchnorm, but no results regarding computational time have been reported. Under the low-precision setting, L1 is shown to have much better convergence compared to L2 normalization. However, the experimental setting is not documented (e.g., the dataset, model, hyperparameters). It is unclear whether L2 BN could work by tuning related hyperparams such as the batch size and the learning rate. It is also not clear what baseline is presented in Figure 3 and how it is compared to state-of-the-art performance. More importantly, all claims made in Section 4 are only supported by at most one experiment on one dataset without comparison to any state-of-the-art results, which poses questions on the generalization of these phenomena. Second, Section 5 is below the standard of NIPS both in terms of novelty and significance. Fixing the scale of weight norm is pretty straightforward, and it is worse than the batchnorm. Third, the paper is loosely organized. The three pieces of analysis and results seem unrelated. The authors tempt to glue them by telling a story of viewing batchnorm as decoupling weight scales and directions, but I don't think weight decay and weight norm should really be relevant. I would suggest that in the later version of the paper, the authors should focus on the L1 normalization part of the paper and improve the experiments by considering a much more diverse setting. Update: The authors' response addressed some of my concerns. I still feel that the scope of the experiments is somewhat limited, but I agree with the other reviewers that the paper brings insight and has potential. And thus I raised my score to vote for acceptance.